# Probing the Robustness of Independent Mechanism Analysis
# for Representation Learning

**Joanna Sliwa**[1,2]    **Shubhangi Ghosh**[1,3]    **Vincent Stimper**[1,4]    **Luigi Gresele**[1]    **Bernhard Schölkopf**[1,3]

[1]Max Planck Institute for Intelligent Systems, Tübingen, Germany
[2]Eberhard Karls University of Tübingen, Tübingen, Germany
[3]Swiss Federal Institute of Technology, Zürich, Switzerland
[4]University of Cambridge, Cambridge, United Kingdom

## Abstract

One aim of representation learning is to recover the original latent code that generated the data, a task which requires additional information or inductive biases. A recently proposed approach termed Independent Mechanism Analysis (IMA) postulates that each latent source should influence the observed mixtures *independently*, complementing standard nonlinear independent component analysis, and taking inspiration from the principle of independent causal mechanisms. While it was shown in theory and experiments that IMA helps recovering the true latents, the method's performance was so far only characterized when the modeling assumptions are exactly satisfied. Here, we test the method's robustness to violations of the underlying assumptions. We find that the benefits of IMA-based regularization for recovering the true sources extend to mixing functions with various degrees of violation of the IMA principle, while standard regularizers do not provide the same merits. Moreover, we show that unregularized maximum likelihood recovers mixing functions which systematically deviate from the IMA principle, and provide an argument elucidating the benefits of IMA-based regularization.

## 1 INTRODUCTION

One objective of representation learning is to invert the data generating process, recovering the underlying factors of variation which generated the observations [Bengio et al., 2013]. A closely related objective is blind source separation (BSS) [Jutten and Hérault, 1991]: given measurements which are mixtures of some latent sources, the aim is to recover them up to tolerable ambiguities. A method to solve BSS is Independent Component Analysis (ICA) [Comon,

1994], under the additional assumption that the sources are statistically independent. If the mixing is nonlinear, however, the model is nonidentifiable without additional constraints, i.e. the reconstructed sources, even though independent, might not be the true ones. Recently, Gresele et al. [2021] proposed a new method called Independent Mechanism Analysis (IMA) to address this problem. This extends Independent Component Analysis by additionally requiring that the sources should influence the observations *independently*, where independence is meant in a nonstatistical sense inspired by the principle of Independent Causal Mechanisms [Peters et al., 2017], thereby providing a causally motivated inductive bias for representation learning. Similar objectives have subsequently been used in the context of generative adversarial networks [Wei et al., 2021] and Principal Manifold Flows [Cunningham et al., 2022].

Gresele et al. [2021] showed that IMA allows to rule out many of the counterexamples (or *spurious solutions*) typically used to show nonidentifiability of nonlinear ICA; and that a regularised likelihood objective based on IMA helps recovering the ground truth sources. However, this approach has so far only been tested on problems where the mixing satisfies the IMA assumption, but for many practical problems of interest, the assumption will most likely not hold exactly, and it is unclear whether the method is robust to deviations from it. Moreover, it is a priori unclear whether the observed benefits of the IMA-regularised likelihood would also be given by other kinds of regularization of the mixing function class—for example, by enforcing that the reconstructed mixing has low complexity or is close to linear [Zhang and Chan, 2008]—or whether they are specific to the IMA principle.

In this work, we aim to close this gap by considering more generic mixing functions, i.e. multilayer perceptrons (MLPs) of varying depth and with randomly sampled parameters, which are frequently used as ground truth mixing functions in the literature [Hyvärinen and Morioka, 2016, Hyvarinen and Morioka, 2017, Khemakhem et al., 2020]: crucially, these are not specifically designed to satisfy the IMA prin-

*Accepted for the Causal Representation Learning workshop at the 38[th] Conference on Uncertainty in Artificial Intelligence* (UAI CRL 2022).

ciple. We quantify to what degree these functions deviate from the IMA assumptions, and investigate whether the IMA contrast still enables us to distinguish between the true and spurious solutions. Furthermore, we compare IMA-based regularization to other types of regularization, investigating whether its benefits are specific to it. We find that the spurious solutions can be ruled out and the ground truth sources reconstructed even when the assumptions are not exactly satisfied, thus indicating a degree of robustness of the method, and that the benefits of IMA-based regularization are not observed for other standard regularizers. Moreover, we find that unregularized maximum likelihood systematically deviates from the IMA principle, and provide an argument to explain why this happens and how different levels of IMA-based regularization can be beneficial.

## 2 BACKGROUND

### 2.1 INDEPENDENT COMPONENT ANALYSIS AND ESTIMATION

Independent Component Analysis (ICA) is one approach to solve BSS [Comon, 1994]. It assumes ground truth latent sources $\mathbf{s} \in \mathbb{R}^n$ and observations $\mathbf{x} \in \mathbb{R}^n$. The relation between $\mathbf{s}$ and $\mathbf{x}$ is an invertible transformation

$$\mathbf{x} = \mathbf{f}(\mathbf{s}),$$

where $\mathbf{f}$ is also termed *mixing function*. Moreover, it assumes that the sources are statistically independent, $p_\mathbf{s}(\mathbf{s}) = \prod_{i=1}^n p_{s_i}(s_i)$. The task of independent component analysis is to transform $\mathbf{x}$ into independent components, i.e. we need to find a transformation $\mathbf{g} : \mathbb{R}^n \to \mathbb{R}^n$, $\mathbf{y} = \mathbf{g}(\mathbf{x})$, which should result in the estimated components $y_i$ being statistically independent. Such a mapping $\mathbf{g}$ can for example be found through maximum likelihood estimation (MLE), where the likelihood $\mathcal{L}(\theta; \mathbf{x})$ is maximized for some class of models parametrized by $\theta$. This provides a way to estimate the parameters $\hat{\theta}$ of $\mathbf{g}$, and can be written as follows:

$$\hat{\theta} = \arg \max_\theta \mathcal{L}(\theta; \mathbf{x}).$$

Whether the estimated independent components $y_i$ recover (or "separate") the ground truth sources $s_i$ depends on a property of the model $(\mathbf{f}, p_\mathbf{s})$ called identifiability.

### 2.2 IDENTIFIABILITY

Here we adopt the notation from [Gresele et al., 2021]. Let $\mathcal{F}$ be the space of all smooth, invertible functions $\mathbf{f} : \mathbb{R}^n \to \mathbb{R}^n$ and let $\mathcal{P}$ be the space of all smooth, factorized densities, $p_\mathbf{s}$ on $\mathbb{R}^n$. Further let $\mathcal{M} \subseteq \mathcal{F} \times \mathcal{P}$ be a subspace of models and let $\sim$ be an equivalence relation on $\mathcal{M}$. Denote by $\mathbf{f}_* p_\mathbf{s}$ the push-forward density of $p_\mathbf{s}$ via $\mathbf{f}$. Then, the generative model is said to be $\sim$-identifiable on $\mathcal{M}$ if:

$$\forall (\mathbf{f}, p_\mathbf{s}), (\bar{\mathbf{f}}, p_{\bar{\mathbf{s}}}) \in \mathcal{M} :$$

$$\mathbf{f}_* p_\mathbf{s} = \bar{\mathbf{f}}_* p_{\bar{\mathbf{s}}} \quad \implies \quad (\mathbf{f}, p_\mathbf{s}) \sim (\bar{\mathbf{f}}, p_{\bar{\mathbf{s}}})$$

Intuitively, it means that the ground truth sources may be reconstructed up to some ambiguities specified by the equivalence relation "$\sim$". For example, identifiablity of linear ICA has been analyzed in [Comon, 1994], where it was shown that the true sources can be recovered up to permutation and rescaling provided at most one of the true latents is Gaussian. When $\mathbf{f}$ is nonlinear, however, the model is in general non-identifiable: one can construct maps which yield independent components while not solving BSS. An example of this is given by the Darmois construction $\mathbf{g}^{\mathbf{D}} : \mathbb{R}^n \to (0, 1)^n$ [Hyvärinen and Pajunen, 1999]:

$$g_i^{\mathrm{D}}(\mathbf{x}_{1:i}) = \int_{-\infty}^{x_i} p(x_i' | \mathbf{x}_{1:i-1}) dx_i'$$

That is, the construction recursively applies the conditional Cumulative Distribution Function (CDF) transform. The Jacobian of the resulting map $\mathbf{g}^{\mathbf{D}}$ is lower-triangular. The reconstructed sources using such transformation will be independent but will in general not solve BSS. Identifiability results for nonlinear ICA can be given for settings where an auxiliary variable $\mathbf{u}$ (e.g., an environment index, time stamp, class label) renders the sources *conditionally independent* [Hyvärinen et al., 2019, Gresele et al., 2019, Khemakhem et al., 2020, Hälvä and Hyvärinen, 2020].

### 2.3 INDEPENDENT MECHANISM ANALYSIS

To deal with the case where no auxiliary variables are available, Gresele et al. [2021] propose an approach termed Independent Mechanism Analysis (IMA). They take inspiration from the principle Independent Causal Mechanisms:

**Principle 1** (Independent Causal Mechanisms [Peters et al., 2017]). *The causal generative process of a system's variables is composed of autonomous modules that do not inform or influence each other.*

In Principle 1, independence is meant in the nonstatistical sense of "no fine tuning" between the causal mechanisms; various formalisations of this principle exist [Janzing and Schölkopf, 2010, Janzing et al., 2012, Besserve et al., 2018].

In IMA, a similar "nonstatistical" independence is assumed over the influences of the individual sources on the observations. The key assumption is that the contributions of different sources $s_i$ to the observations through the mixing function $\mathbf{f}$ are independent. This can be formalized as follows:

$$\log |\mathbf{J}_\mathbf{f}(\mathbf{s})| = \sum_{i=1}^n \log \left|\left| \frac{\partial \mathbf{f}}{\partial s_i}(\mathbf{s}) \right|\right|,$$

i.e. the contributions $\partial \mathbf{f}/\partial s_i$ from each source to the mixing mechanisms (which are the columns of the Jacobian $\mathbf{J_f}$) are orthogonal. The authors of the paper introduce a function to measure the IMA principle and show how it can be useful in nonlinear BSS. This is called global IMA contrast and is given by:

$$C_{\mathrm{IMA}}(\mathbf{f}, p_s) = \mathbb{E}_{\mathbf{s} \sim p_s} \left[ \sum_{i=1}^{n} \log \left\| \frac{\partial \mathbf{f}}{\partial s_i}(\mathbf{s}) \right\| - \log |\mathbf{J_f}(\mathbf{s})| \right]$$

It quantifies how the IMA principle is violated for a solution $(\mathbf{f}, p_\mathbf{s})$. The properties of $C_{\mathrm{IMA}}$ are as follows:

1. $C_{\mathrm{IMA}}(\mathbf{f}, p_\mathbf{s}) \geq 0$, and equal only when $\mathbf{J_f} = \mathbf{O}(\mathbf{s})\mathbf{D}(\mathbf{s})$ where $\mathbf{O}(\mathbf{s})$ is an orthogonal matrix and $\mathbf{D}(\mathbf{s})$ a diagonal matrix,

2. $C_{\mathrm{IMA}}(\mathbf{f}, p_\mathbf{s}) = C_{\mathrm{IMA}}(\tilde{\mathbf{f}}, p_{\tilde{\mathbf{s}}})$ for $\tilde{\mathbf{f}} = \mathbf{f} \circ \mathbf{h}^{-1} \circ \mathbf{P}^{-1}$ and $\tilde{\mathbf{s}} = \mathbf{Ph}(\mathbf{s})$ where $\mathbf{P}$ is a permutation and $\mathbf{h}$ an invertible element-wise function.

The authors show that Darmois construction solutions have strictly positive $C_{\mathrm{IMA}}$. For a mixing function where $C_{\mathrm{IMA}} = 0$, it is therefore possible to distinguish the Darmois construction from the true one; similarly, $C_{\mathrm{IMA}}$ distinguishes other common classes of spurious solutions introduced by [Locatello et al., 2019] from the true ones. Moreover, $C_{\mathrm{IMA}}$ is blind to permutation and element-wise transformation of the sources, which are unresolvable ambiguities of nonlinear ICA [Hyvärinen et al., 2001].

## 3 EXPERIMENTS

In the following experiments, firstly we define a mapping that we use as a model of generic mixing functions occurring in practical applications. Then, we compute the IMA contrast and observe what happens when such mixing deviates from the assumptions of IMA and if we can still distinguish between true and spurious solutions. For that mixing, deviating from the method's assumptions, we check how good the reconstruction of the sources is using $C_{\mathrm{IMA}}$-regularized MLE. Moreover, we compare regularization with the $C_{\mathrm{IMA}}$ with other types of regularization.

### 3.1 GENERIC MIXING FUNCTIONS

Firstly, we define a framework to model and generate generic mixing functions. In the literature [Hyvärinen and Morioka, 2016, Khemakhem et al., 2020, Hyvarinen and Morioka, 2017], authors have used MLPs since they offer a nonlinear, complex mixing that have an increasing complexity the more layers are used and they can resemble real-world maps. Therefore, our setting is as follows:

1. **Mixing.** The mixing function is an MLP with `leaky_tanh` [Gresele et al., 2020] as the activation.

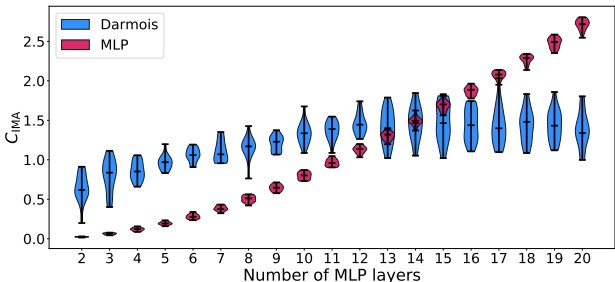

Figure 1: $C_{\mathrm{IMA}}$ of randomly initialized MLPs with varying number of layers and the corresponding Darmois construction.

The weights for each layer are initialized as orthogonal matrices and the biases are normally-distributed random arrays. The invertibility of the mixing function is ensured as the activation function and the weight matrices are invertible—and so is the composition of them. We check results for an increasing number of layers $L \in \{2, 3, \ldots, 20\}$. The dimensionality of the data is $n = 5$.

2. **Spurious solution.** To learn the spurious solutions, i.e. to estimate the Darmois construction for our case, we use residual normalizing flows with triangular Jacobian [Gresele et al., 2021] with a Gaussian base distribution. We maximize the likelihood over 100 000 iterations for 20 different mixings.

For such mixing functions, we want to check if the generic map still satisfies the assumptions of IMA, i.e. that the columns of the Jacobian are orthogonal, corresponding to $C_{\mathrm{IMA}} = 0$. Figure 1 shows the $C_{\mathrm{IMA}}$ values for the true mixing (MLP) and for the spurious solution (Darmois construction) for an increasing number of layers. The more layers we use, the further the mixing deviates from the IMA assumption. Note that linear orthogonal transformations satisfy IMA, but maps which intertwine them with elementwise nonlinearities do not necessary do so. Apparently, errors seem to accumulate, i.e. the deviation from the IMA principle grows monotonically in $L$. However, we notice that up to 10-12 layers spurious solutions possess higher $C_{\mathrm{IMA}}$ than the true mixing functions. Therefore, the true mixing is distinguishable up to some point and IMA appears to be useful even for such mixings. In related works [Hyvärinen and Morioka, 2016, Khemakhem et al., 2020, Hyvarinen and Morioka, 2017], MLPs with up to 5 layers are used, meaning that the $C_{\mathrm{IMA}}$ could still be useful there.

The results shown here and in the later experiments were, however, obtained using orthogonal weights initialization, whereas in the literature [Hyvärinen and Morioka, 2016] the initialization was uniform. Results for such initialization show a less clear separation between true and spurious solutions and can be found in the Appendix A.

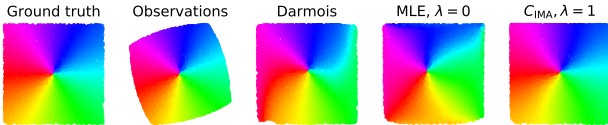

Figure 2: Visualization of the ground truth sources and the observations generated by an 4-layered MLP as well as the reconstructed sources using the Darmois construction, MLE, and MLE with $C_{\text{IMA}}$-regularization.

## 3.2 QUALITY OF SOURCE RECONSTRUCTION

Since even for some mixing functions that deviate from IMA assumption the method appears to identify spurious solutions, we want to investigate whether we can use the IMA contrast to solve BSS. In order to do so, we use the objective function from [Gresele et al., 2021]. The authors propose a maximum likelihood approach with $C_{\text{IMA}}$-regularization:

$$\mathcal{L}(\mathbf{g}; \mathbf{x}) = \mathbb{E}_{\mathbf{x}}[\log p_{\mathbf{g}}(\mathbf{x}) - \lambda C_{\text{IMA}}(\mathbf{g}^{-1}, p_{\mathbf{y}})]$$

where $\mathbf{g}$ is the learnt unmixing, $\mathbf{y}$ the reconstructed sources, $\lambda$ is the Lagrange multiplier and it determines how big the regularization is ($\lambda = 0$ is just MLE). Ideally, we would like our model to solve BSS. The setting for the following experiments is:

1. **Mixing.** The mixing function we use is again an MLP mixing function. The dimensionality of the data is $n = \{2, 5\}$ and number of layers $L \in \{2, 3, \ldots, 20\}$. The other parameters are the same as before.

2. **Learnt unmixing.** To learn the unmixing, we use now residual normalizing flows with full Jacobian [Chen et al., 2019] and the base distribution is changed to a logistic distribution. We maximise the likelihood of the data with $C_{\text{IMA}}$-regularization for $\lambda \in \{0, 0.5, 1.0\}$ over 100 000 iterations for 20 different mixing functions.

### 3.2.1 Visualization

Firstly, we want to inspect whether the Darmois construction, MLE and $C_{\text{IMA}}$-regularized MLE recovers the mixing visually the best for a two-dimensional mixing function. In Figure 2, we can assess that MLE $\lambda = 1$ gives the best results for a 4-layered MLP. It produces sources closer to the true sources than Darmois construction or unregularized MLE ($\lambda = 0$), although some slight distortions are visible. Results for more or less layers can be found in the Appendix A.

### 3.2.2 Metrics

To quantify how close the reconstructed sources are to the ground truth we use the mean correlation coefficient (MCC).

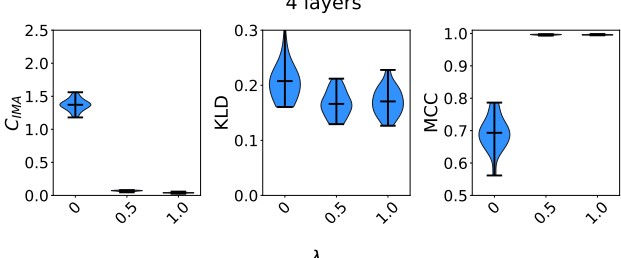

Figure 3: $C_{\text{IMA}}$, KLD to the ground truth, and MCC of flow models trained with various levels of $C_{\text{IMA}}$-regularization on data generated by an MLP with 4 layers.

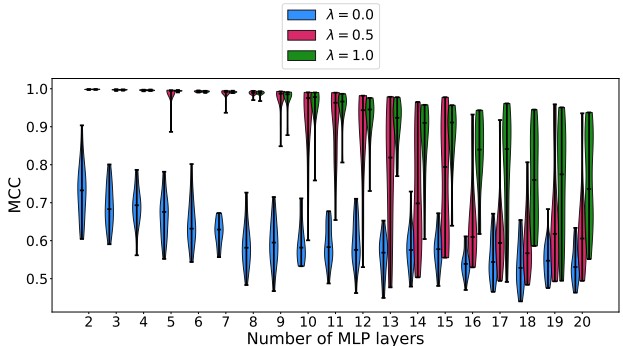

Figure 4: MCC of flow models trained with various levels of $C_{\text{IMA}}$-regularization on data generated by an MLP with varying number of layers.

Higher values mean that the reconstruction is closer to the true sources. We can check the MCC between the original sources and the corresponding latents. We compute the Spearman correlation matrix of the true and reconstructed sources and after that we need to match them where they have the highest correlation with the Hungarian algorithm.

In Figure 3 we show the value of the contrast function for different values of regularization as well as the KLD and MCC for 5-dimensional MLPs with 4 layers. We can notice that with bigger values of $\lambda$, $C_{\text{IMA}}$ is decreasing. MCC gets bigger with higher regularization. Regarding the fit of the model to the data, the KLD is low across all values of regularization; however, we can observe a small decrease when regularization is applied. Metrics for different number of layers can be found in the Appendix A.

Next, we want to check what happens to MCC value across different number of layers of 5-dimensional MLPs for unregularized and regularized models. In Figure 4 we can see that MCC gets worse with increasing number of layers. Bigger $C_{\text{IMA}}$-regularization results in higher values of MCC compared to unregularized results. After around 10 layers the distributions of the results are overlapping, but on average higher values of $\lambda$ still lead to higher MCC mean values.

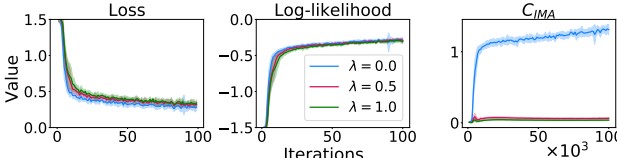

Figure 5: Loss, log-likelihood and $C_{\text{IMA}}$ values across training for a 4-layered MLP and the dimensionality of the data $n = 5$.

Note also that whereas for $\lambda = 0$ the variance in MCC values is high across all values of $L$, for $\lambda \in \{0.5, 1\}$ it tends to be low (and the mean tends to concentrate near 1) for lower $L$, and high for higher $L$. This seems to suggest that for small $L$ (i.e., less violation of the IMA principle) most solutions achieve source separation, whereas as $L$ increases (i.e., more violation of the IMA principle) other solutions which do not separate the sources are found by stochastic gradient descent: the broader spread in MCC values seems to reflect a larger variety of solutions found when optimizing the $C_{\text{IMA}}$-regularized objective for large $L$.

### 3.3 $C_{\text{IMA}}$ ACROSS TRAINING

In this section, we investigate how the $C_{\text{IMA}}$ changes during training. In Figure 5 we notice that for an unregularized model, i.e. $\lambda = 0$, the $C_{\text{IMA}}$ increases with more iterations and as the log-likelihood increases. In constrast, for regularized models instead, i.e. $\lambda \in \{0.5, 1\}$, the $C_{\text{IMA}}$ does not grow significantly. Note that the log likelihood increases for all values of the regularization. A similar figure for two dimensions can be found in the Appendix A, as well as more in depth theoretical justification for the behavior of $C_{\text{IMA}}$ during training in this setting.

We can get an intuition on the observed behavior by writing the IMA-regularized likelihood for a single point in $n = 2$,

$$\mathcal{L}(\mathbf{g}; \mathbf{x}) = \underbrace{\log p_{\mathbf{y}}(\mathbf{y})}_{(i)} - \underbrace{(\log \|\mathbf{a}\| + \log \|\mathbf{b}\|)}_{(ii)}$$
$$- (1 - \lambda) \underbrace{\log |\sin \theta|}_{(iii)} \tag{1}$$

where $\mathbf{y} = \mathbf{g}(\mathbf{x})$ are the learned sources, $\mathbf{a}, \mathbf{b}$ are the columns of the Jacobian evaluated at $\mathbf{y}$ and $\theta$ is the angle between them, see Appendix B for details. For unregularized maximum likelihood ($\lambda = 0$), maximization of (1) can be achieved by minimising (iii), leading to more collinear columns. We show in Figure 14, that the gradient of this term is steep close to $\theta = 0$, suggesting that it may dominate maximum likelihood estimation, as empirically observed in our experiments. In contrast, the term in (ii) may have the opposite effect, i.e. encouraging column orthogonality. At the optimal value of (i), the area element given by the

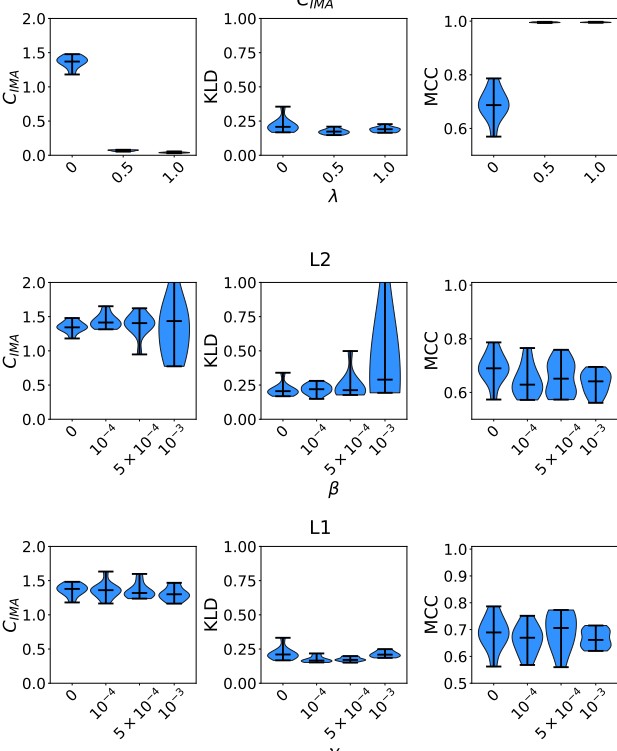

Figure 6: $C_{\text{IMA}}$, KLD, and MCC of the models trained on data generated by a 4-layered MLP with $C_{\text{IMA}}$-, L1-, or L2-regularization applied.

Jacobian determinant $\|\mathbf{a}\| \|\mathbf{b}\| |\sin \theta|$ is fixed[1], and maximization of (1) amounts to minimization of (ii), which for fixed area of the parallelogram spanned by $\mathbf{a}, \mathbf{b}$ is achieved when the two columns are orthogonal. This suggests that for $0 < \lambda < 1$, orthogonality is encouraged by down-weighing the term in (iii), whereas for $\lambda = 1$, (iii) vanishes and optimisation of (ii) yields orthogonal columns at the optimum. Related hypotheses on the behavior of unregularized MLE were reported in Cunningham et al. [2022]; the empirical evidence and theoretical insights that unregularized maximum likelihood training systematically deviates from the IMA principle provide additional arguments in this regard, elucidating the benefits of explicit IMA regularization.

### 3.4 COMPARISON TO OTHER REGULARIZATION TYPES

In the previous sections, we demonstrated that regularization with the $C_{\text{IMA}}$ is a useful tool in representation learning. Here, we compare this approach to other regularization methods, namely L1- [Santosa and Symes, 1986, Tibshirani, 1996] and L2-regularization [Hoerl and Kennard, 1970],

---

[1]It can be shown that any two models optimally fitting the data will achieve the optimal value of (i) and have equal area elements, see Appendix B for details.

which are given by

$$\mathcal{L}(\mathbf{g}; \mathbf{x}) = \mathbb{E}_\mathbf{x}[\log p_\mathbf{g}(\mathbf{x})] - \gamma \sum |\theta_i|,$$

$$\mathcal{L}(\mathbf{g}; \mathbf{x}) = \mathbb{E}_\mathbf{x}[\log p_\mathbf{g}(\mathbf{x})] - \beta \sum_i \theta_i^2,$$

where $\theta_i$ are the weight parameters of the model, not including the biases. We considered the following setting:

1. **Mixing.** The mixing functions we use is again MLPs. The dimensionality of the data is $n = 5$ and the number of layers is $L = 4$. Other parameters are the same as before.

2. **Learnt unmixing.** To learn the unmixing, we use the same residual normalizing flows architecture. We train models for 10 different mixings for each regularization. We maximise the likelihood of the data with L1- and L2-regularization for $\beta, \gamma \in \{0, 10^{-4}, 5 \times 10^{-4}, 10^{-3}\}$.

Figure 6 shows a quantitative comparison of the regularization types. When applying L1- or L2-regularization, the $C_{\mathrm{IMA}}$ of the learned models remains unchanged and the KLD is mostly in a similar range for all regularization types. In terms of the MCC, neither increased L1- nor L2-regularization leads to improvement, while regularizing with the $C_{\mathrm{IMA}}$ boosts the performance significantly. Hence, $C_{\mathrm{IMA}}$-regularization should be preferred over the traditional techniques.

## 4 CONCLUSION

We use randomly initialised MLPs as a generic nonlinear mixing which is not a priori designed to satisfy the IMA principle. For such mixings, we notice that with the increase of the number of layers, the IMA principle is increasingly violated; nevertheless, $C_{\mathrm{IMA}}$ still allows distinguishing true and spurious solutions for a broad range of cases. Additionally, $C_{\mathrm{IMA}}$-regularized MLE approach produces better source reconstruction than the typical MLE. Finally, the benefits of IMA regularization are not matched by other more standard regularizers. Overall, our results indicate that IMA may be a useful method for nonlinear BSS even when the ground truth mixing violates the IMA principle to some extent. This suggests that the approach could be a useful tool for representation learning even in more realistic settings where the modeling assumptions are not exactly satisfied.

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

# A ADDITIONAL RESULTS

Here, we present some additional results which extend the scope of the experiments from Section 3.

**Fit of the model for spurious solutions.** In the setting from Figure 1, we want to check how well the model learns the spurious solutions (Darmois construction) across different number of layers. In Figure 7, we can notice that KLD has higher values with more layers. This means that the goodness of fit of the Darmois construction solutions to the data is getting worse. This result is intuitive as the Darmois construction for a mixing function with more layers is harder to learn. A possible solution could be to extend the time of the training or add more layers in the normalizing flows architecture for higher values of $L$.

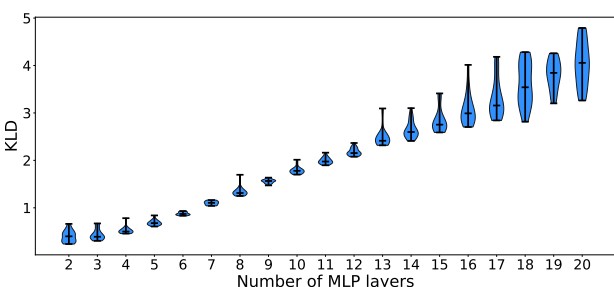

Figure 7: KLD across layers for a spurious solution, i.e. the Darmois construction.

**Uniform initialization of MLP weights.** The results from Section 3 have been obtained using orthogonal weights initialization. Here, we want to investigate if the spurious solutions are still distinguishable from the true mixing as in Figure 1 if the initialization is as in the literature [Hyvärinen and Morioka, 2016]. The initialization for the weights $\theta$ is as follows :

$$\theta \sim U\left[-\frac{1}{\sqrt{n}}, \frac{1}{\sqrt{n}}\right]$$

where $n$ is the size of the layer (the number of columns of $\theta$) and $U[-a, a]$ is a uniform distribution in the interval $(-a, a)$. Biases were set to zeros. In Figure 8 we can observe that for such initialization with layers $L \in \{2, 3, 4, 5\}$, the spurious solutions obtain lower values of $C_{\text{IMA}}$ than the true mixing. Therefore, it is impossible to distinguish between them and choose the true mixing. This result highlights the limitation of the method for MLP mixings based on the initialization.

**Visualization of source reconstruction for less and more layers.** Next, we visually inspect the reconstructed sources using three different methods: Darmois construction, MLE and $C_{\text{IMA}}$-regularized MLE with $\lambda = 1$ for a MLP mixing where $L = 2$ and $L = 8$. This enables us to see how the methods perform for less and more complex mixings than the one used in Figure 2. In Figure 9 we can notice that for both 2 and 8 layers $C_{\text{IMA}}$-regularized MLE performs better

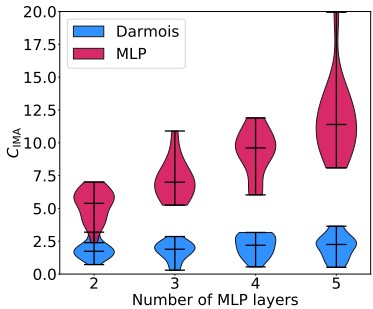

Figure 8: $C_{\text{IMA}}$ of uniformly initialized MLPs with with varying number of layers and the corresponding Darmois construction.

than Darmois construction or unregularized MLE. For a more complex mixing, we can observe that in the $C_{\text{IMA}}$-regularized MLE reconstructed source, slight distortions are present.

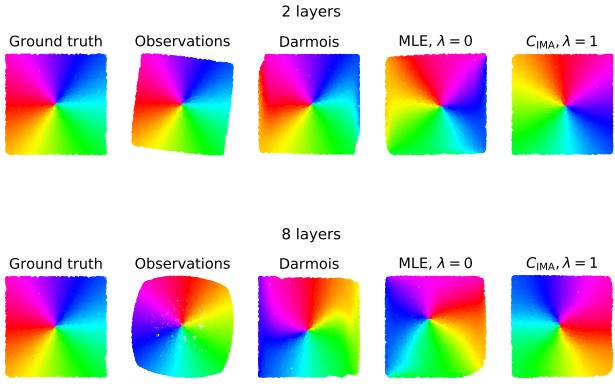

Figure 9: Visualization of the ground truth sources and the observations generated by an (*top*) 2-layered and (*bottom*) 8-layered MLP as well as the reconstructed sources using the Darmois construction, MLE, and MLE with $C_{\text{IMA}}$-regularization.

**Metrics for less and more layers across different regularization values.** Following the previous results, now we would like to quantitatively check how close the reconstructed sources are to the ground truth for less and more complex mixings. As in Figure 3 we show the value of the contrast function for different values of regularization as well as the KLD and MCC. In Figures 10 & 11, we can see that for less layers, KLD mean seems to be decreasing for increasing $\lambda$ value. For more layers, it seems to stay at the same level. Other metrics show the same trend as for 4 layers - lower $C_{\text{IMA}}$ and higher MCC with regularization.

$C_{\text{IMA}}$ **across training for dimensionality** $n = 2$. In Section 3.3, we showed in Figure 5 the results for $n = 5$. Now, we want to observe the training behavior for $n = 2$. In Figure 12 we can see the loss value, log-likelihood and $C_{\text{IMA}}$ across training. For $\lambda = 1$, $C_{\text{IMA}}$ increases at a quicker pace than

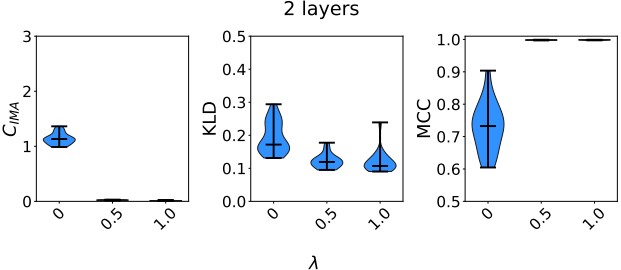

Figure 10: $C_{\text{IMA}}$, KLD to the ground truth, and MCC of flow models trained with various levels of $C_{\text{IMA}}$-regularization on data generated by an MLP with 2 layers.

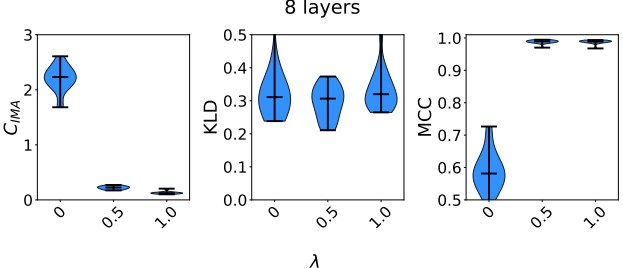

Figure 11: $C_{\text{IMA}}$, KLD to the ground truth, and MCC of flow models trained with various levels of $C_{\text{IMA}}$-regularization on data generated by an MLP with 8 layers.

for $n = 5$ until the moment log-likelihood is increasing and then $C_{\text{IMA}}$ stays at the same level.

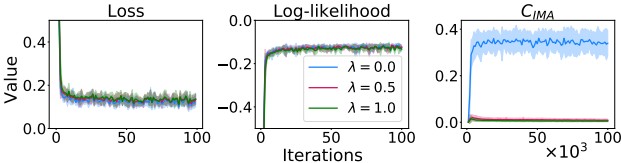

Figure 12: Loss, log-likelihood and $C_{\text{IMA}}$ values across training for a 4-layered MLP and the dimensionality of the data $n = 2$.

# B $C_{\text{IMA}}$ ACROSS TRAINING - THEORETICAL INSIGHTS

Next, we elaborate on the theoretical intuition for the empirical observation that $C_{\text{IMA}}$ grows in unregularized maximum likelihood training as show in Section 3.3.

**Geometric interpretation of $C_{\text{IMA}}$-regularlized likelihood in 2-D**

We consider a simplified setting, the dimensionality of the data is $n = 2$. We write the $C_{\text{IMA}}$-regularized likelihood at a point $\mathbf{x} = \mathbf{f}(\mathbf{s})$, in terms of the norms of the columns of the Jacobian, $\mathbf{J}_{\mathbf{g}^{-1}}(\mathbf{y})$, and the angle between them, where

$\mathbf{g}$ is the learned unmixing and $\mathbf{y}$ are the learned sources.

Let $\mathbf{a}, \mathbf{b}$ be the columns of $\mathbf{J}_{\mathbf{g}^{-1}}(\mathbf{y})$ and $\theta$ be the angle between them. The determinant of $\mathbf{J}_{\mathbf{g}^{-1}}(\mathbf{y})$ is given by the area of the parallelogram spanned by $\mathbf{a}$ and $\mathbf{b}$, $|\mathbf{J}_{\mathbf{g}^{-1}}(\mathbf{y})| = \|\mathbf{a}\|\|\mathbf{b}\| \sin \theta^2$.

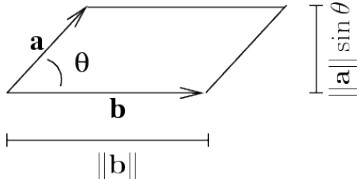

Figure 13: Area of parallelogram.

Using this fact, we parse the $C_{\text{IMA}}$-regularized log likelihood at a point $\mathbf{x}$.

$$\mathcal{L}(\mathbf{g}; \mathbf{x}) = \log p_{\mathbf{g}}(\mathbf{x}) - \lambda C_{\text{IMA}}(\mathbf{g}^{-1}, p_{\mathbf{y}}),$$

where $p_{\mathbf{y}}$ is the chosen base distribution.

$$
\begin{aligned}
\mathcal{L}(\mathbf{g}; \mathbf{x}) &= \log p_{\text{g}}(\mathbf{x}) - \lambda C_{\text{IMA}}(\mathbf{g}^{-1}, p_{\text{y}}) \\
&= \log p_{\text{y}}(\mathbf{y}) - \log \left|\mathbf{J}_{\mathbf{g}^{-1}}(\mathbf{y})\right| - \lambda C_{\text{IMA}}(\mathbf{g}^{-1}, p_{\text{y}}) \\
&= \log p_{\text{y}}(\mathbf{y}) - \log \left|\mathbf{J}_{\mathbf{g}^{-1}}(\mathbf{y})\right| \\
&\quad - \lambda \left(\log \|\mathbf{a}\| + \log \|\mathbf{b}\| - \log \left|\mathbf{J}_{\mathbf{g}^{-1}}(\mathbf{y})\right|\right) \\
&= \log p_{\text{y}}(\mathbf{y}) - \log \left|\|\mathbf{a}\|\|\mathbf{b}\| \sin \theta\right| \\
&\quad - \lambda \left(\log \|\mathbf{a}\| + \log \|\mathbf{b}\| - \log \left|\|\mathbf{a}\|\|\mathbf{b}\| \sin \theta\right|\right)
\end{aligned}
$$

Hence, we show (1),

$$
\mathcal{L}(\mathbf{g}; \mathbf{x}) = \underbrace{\log p_{\text{y}}(\mathbf{y})}_{\text{(i)}} - \left(\underbrace{\log \|\mathbf{a}\| + \log \|\mathbf{b}\|}_{\text{(ii)}}\right) \\
- (1 - \lambda) \underbrace{\log |\sin \theta|}_{\text{(iii)}} \tag{1}
$$

## $C_{\text{IMA}}$ across training

In the case of unregularized maximum likelihood training ($\lambda = 0$), maximizing $\mathcal{L}(\mathbf{g}; \mathbf{x})$ leads to minimizing all the terms in (1)—and in particular both terms that involve the Jacobian of the learned unmixing, (ii) and (iii). Clearly, lower values of (iii) promote functions which have collinear columns in the Jacobian. In contrast, minimizing the term in (ii) may have the opposite effect i. e. encouraging column orthogonality. To show this, we compare the loss for learned unmixings which have the same value of likelihood, $\log p_{\mathbf{x}}(\mathbf{x})$. See below that, at the optimal value for the

---

[2]https://proofwiki.org/wiki/Area_of_Parallelogram_from_Determinant

learned latent likelihood $p_\mathbf{y}(\mathbf{y})$, i. e. for fixed (i), this results in comparing the loss for same value of area element given by the determinant of the Jacobian. Consider two models, given by the learned unmixing functions $\mathbf{g}^{(1)}$ and $\mathbf{g}^{(2)}$. For $\mathbf{y}^{(1)} = [\mathbf{g}^{(1)}]^{-1}(\mathbf{x})$, $\mathbf{y}^{(2)} = [\mathbf{g}^{(2)}]^{-1}(\mathbf{x})$: we compare the loss for two such models with the same value of likelihood, and optimal latent likelihood, $p_\mathbf{y}(\mathbf{y}^{(1)}) = p_\mathbf{y}(\mathbf{y}^{(2)})$,

$$
\begin{aligned}
\log p_\mathbf{x}(\mathbf{x}) &= \log p_\mathbf{y}(\mathbf{y}^{(1)}) - \log |\mathbf{J}_{[\mathbf{g}^{(1)}]^{-1}}(\mathbf{y}^{(1)})| \\
&= \log p_\mathbf{y}(\mathbf{y}^{(2)}) - \log |\mathbf{J}_{[\mathbf{g}^{(2)}]^{-1}}(\mathbf{y}^{(2)})| \\
\implies |\mathbf{J}_{[\mathbf{g}^{(1)}]^{-1}}(\mathbf{y}^{(1)})| &= |\mathbf{J}_{[\mathbf{g}^{(2)}]^{-1}}(\mathbf{y}^{(2)})|
\end{aligned}
$$

Among unmixing functions corresponding to the same area element given by the Jacobian determinant, the function which minimizes (ii), is the one which has orthogonal Jacobian columns. This is because, by the Isoperimetric Theorem for parallelograms[3]: among parallelograms with the same area, the one with the minimum perimeter is a rectangle. This supports our claim that minimizing (ii) encourages column orthogonality in the Jacobian.

Given the contrasting effects of optimizing (ii) and (iii), we empirically observe that (iii) dominates maximum likelihood training. A possible justification is the steep gradient of $\log |\sin \theta|$ close to $\theta = 0$, which results in large gradients for the parameters by chain rule of differentiation.

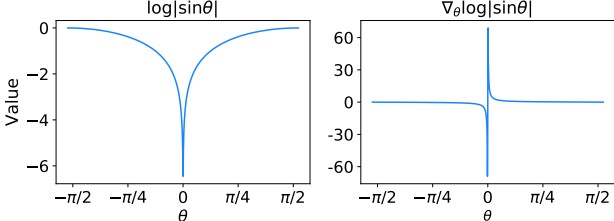

Figure 14: $\log |\sin \theta|$ and its gradient.

Further, our empirical observations suggest that for $0 < \lambda < 1$, orthogonality is encouraged by down-weighing the term in (iii), whereas for $\lambda = 1$, (iii) vanishes and optimisation of (ii) yields orthogonal columns at the optimum.

[3]https://www.cut-the-knot.org/m/Geometry/
ParallelogramToRectangle.shtml
