# OpenReview forum: "Probing the Robustness of Independent Mechanism Analysis for Representation Learning"
_auai.org/UAI/2022/Workshop/CRL — CRL@UAI 2022 Poster_

### Official Review · Reviewer_UXqQ · 2022-06-27
**Well-founded paper providing empirical evaluation of IMA**

**Rating:** 9
**Confidence:** 4

**Review:**

## Summary

The main contribution of this paper is an empirical evaluation and analysis of Independent Mechanism Analysis (IMA) when some of the model's central assumptions are broken. Specifically, when the mixing function consists of randomly initialized MLPs. The main metric reported in the evaluation of the sources is the Mean Correlation Coefficient (MCC), which is standard in ICA,  while the global IMA contrast quantifies how much the IMA principle is violated. Experiments show that IMA regularization is beneficial when compared to other regularizers.
Overall, the contributions are solid and this work is a good addition to the IMA/ICA literature, which often lacks thorough empirical evaluations.

## Pros

- The paper is well-written and introduces the background of IMA.
- The experiments are well-designed and fulfill the purpose of the paper. The number of mixing functions and MLP layers seems appropriate for the experiments.
- The contrast between IMA-regularized models and unregularized models in this particular setup is clear from the plots, not raising doubts about the conclusion.

## Cons

- The dimensionality of the data, n={2,5}, can be slightly limiting since it is not clear that the same conclusions would hold on high-dimensional spaces. However, it is understandable that high-dimensional spaces increase the computational complexity of the experiments.
- The number of data points per bin (which seems to be ”20 different mixing functions”) seems appropriate for most experiments. However, in Figure 4 there is a high variance in the MCC scores and it would be interesting to know why in order to further understand the conclusions.

---

### Official Review · Reviewer_vtfL · 2022-06-29
**Robustness of regularization based on Independent Mechanism Analysis: empirical results**

**Rating:** 7
**Confidence:** 3

**Review:**

This paper mainly focuses on studying the robustness of independent mechanism analysis (IMA) in recovering latent sources as a regularization term. Specifically, it has been shown in two directions: (1) test the performance of IMA-based regularization when the parametric assumptions are not exactly satisfied; (2) compare IMA-based regularization to other types of regularization.

The motivation is clear and of practical significance. In the original paper on IMA, it has been shown that IMA could help the identifiability of latent sources empirically, though no identifiability result has been given. This paper provides strong empirical results to show that IMA-based regularization could indeed help to recover the true sources even when the assumptions are not exactly satisfied, shedding light on potential full identifiability with assumptions weaker than IMA.

Overall, the contribution is sound and inspiring. One potential future work might be further investigating the power of IMA-regularization theoretically. Perhaps it could help the community could discover a weaker assumption for identifiable BSS.

---

### Meta-Review · Program_Chairs · 2022-07-05

**Recommendation:** Accept (Poster)
**Confidence:** 3

**Metareview:**

Both reviewers gave very high scores to this paper, so I recommend acceptance.

---

### Decision · Program_Chairs · 2022-07-06

Accept (Poster)